# The impact of pulse oximetry and Integrated Management of Childhood Illness (IMCI) training on antibiotic prescribing practices in rural Malawi: A mixed-methods study

Fiona Sylvies[1,2]*, Lucy Nyirenda[3], Alden Blair[1], Kimberly Baltzell[1,4]

**1** Institute for Global Health Sciences, University of California San Francisco, San Francisco, CA, United States of America, **2** Tulane University School of Medicine, New Orleans, LA, United States of America, **3** Global AIDS Interfaith Alliance, Mulanje, Malawi, **4** University of California School of Nursing, University of California San Francisco, San Francisco, CA, United States of America

* fsylvies@tulane.edu

**Data Availability Statement:** The anonymized dataset has been uploaded as Supporting Information (Dataset S2).

## Abstract

### Background

The misdiagnosis of non-malarial fever in sub-Saharan Africa has contributed to the significant burden of pediatric pneumonia and the inappropriate use of antibiotics in this region. This study aims to assess the impact of 1) portable pulse oximeters and 2) Integrated Management of Childhood Illness (IMCI) continued education training on the diagnosis and treatment of non-malarial fever amongst pediatric patients being treated by the Global AIDS Interfaith Alliance (GAIA) in rural Malawi.

### Methods

This study involved a logbook review to compare treatment patterns between five GAIA mobile clinics in Mulanje, Malawi during April-June 2019. An intervention study design was employed with four study groups: 1) 2016 control, 2) 2019 control, 3) IMCI-only, and 4) IMCI and pulse oximeter. A total of 3,504 patient logbook records were included based on these inclusion criteria: age under five years, febrile, malaria-negative, and treated during the dry season. A qualitative questionnaire was distributed to the participating GAIA providers. Fisher's Exact Testing and odds ratios were calculated to compare the prescriptive practices between each study group and reported with 95% confidence intervals.

### Results

The pre- and post-exam scores for the providers who participated in the IMCI training showed an increase in content knowledge and understanding (p<0.001). The antibiotic prescription rates in each study group were 75% (2016 control), 85% (2019 control), 84% (IMCI only), and 42% (IMCI + pulse oximeter) (p<0.001). An increase in pneumonia diagnoses was detected for patients who received pulse oximeter evaluation with an oxygen saturation <95% (p<0.001). No significant changes in antibiotic prescribing practices were detected in

**Funding:** This study was funded by the Global AIDS Interfaith Alliance and the Institute of Global Health Sciences of the University of California, San Francisco.

**Competing interests:** The authors have declared that no competing interests exist.

**Abbreviations:** NMF, non-malarial fever; IMCI, Integrated Management of Childhood Illness; GAIA, Global AIDS Interfaith Alliance.

the IMCI-only group (p>0.001). However, provider responses to the qualitative questionnaires indicated alternative benefits of the training including improved illness classification and increased provider confidence.

## Conclusion

Clinics that implemented both the IMCI course and pulse oximeters exhibited a significant decrease in antibiotic prescription rates, thus highlighting the potential of this tool in combatting antibiotic overconsumption in low-resource settings. Enhanced detection of hypoxia in pediatric patients was regarded by clinicians as helpful for identifying pneumonia cases. GAIA staff appreciated the IMCI continued education training, however it did not appear to significantly impact antibiotic prescription rates and/or pneumonia diagnosis.

## Background and significance

While the diagnosis and treatment of malaria has greatly improved due to the advent of rapid diagnostic tests, non-malarial causes of fever still represent a significant burden of disease throughout sub-Saharan Africa [1]. Within this region, non-malarial fever (NMF) is critical to address in Malawi, particularly amongst patients under the age of five [2]. Of the many causes of NMF in Malawi, pneumonia accounts for 13% of deaths amongst children under the age of five, and is thus essential to accurately diagnose and treat [3]. It is estimated that one in five pediatric deaths due to pneumonia could be avoided if providers showed stronger adherence to existing diagnostic guidelines and interventions [4]. Defined as an of the alveoli, pneumonia typically presents with cough, fatigue, chest tightness, fever, sweating, and shortness of breath [5]. While the gold standard of pneumonia diagnosis usually involves the use of chest x-rays and/or sputum tests, the luxury of this diagnostic equipment is rarely afforded to those living in low-resource settings [6].

Not only does the inaccurate diagnosis of pneumonia lead to adverse patient outcomes, but it also poses a major public health concern due to the subsequent overuse of antibiotic prescriptions. Due to the vague presentation of febrile illness in pediatric patients, Malawian providers tend to over-rely on the prescription of antibiotics for all NMF diagnoses, thus furthering the development of antimicrobial resistance [7–10]. To address the issue of inaccurate diagnoses, additional testing is needed to help providers differentiate febrile patients with or without pneumonia, so that they can determine when antibiotics can be safely distributed or withheld. In the context of low-resource settings, this diagnostic methodology should ideally be cost-effective, function independently of electricity, require minimal training, and provide results that are easy to interpret [11].

Two such diagnostic resources include 1) portable pulse oximeters and 2) the Integrated Management of Childhood Illness (IMCI) continued education courses. Despite being a standard tool in most developed healthcare systems, the pulse oximeter is not widely available in Malawian health facilities, a constraint that is especially exacerbated in rural areas [12, 13]. While studies regarding the use of pulse oximeters and/or IMCI continued education courses are expanding in hospital settings, little is known about the impact of implementing either intervention on antibiotic prescriptive rates in alternative sources of health care delivery, such as mobile clinics. Characteristics inherent in mobile clinics, namely their lack of geographical permanence, may increase providers' over-reliance on antibiotics due to concerns of patients'

febrile symptoms worsening during the time it takes for the clinic to return to a given community [10]. As 85% of the Malawian population resides in rural areas, improving the capacity of mobile clinics to diagnose and treat NMF amongst these hard-to-reach communities is crucial [14].

Addressing these issues are in the forefront of the concerns for providers working for the Global AIDS Interfaith Alliance (GAIA) in Mulanje, Malawi. GAIA is a nonprofit organization which aims to increase healthcare accessibility amongst rural communities based in southern Malawi through the use of mobile clinics that rotate between the many villages throughout this region [15]. The mobile clinics operate five days a week, often extending into the weekend, offering comprehensive primary care services and consistent treatment for conditions that heavily plague this region such as malaria, tuberculosis, and HIV, all free of charge for their patients [15]. GAIA mobile clinic providers operate under tremendously challenging conditions, given that each mobile clinic team typically includes one clinical officer, two nurses and one nurse aid, seeing an average of 160 patients per day [15]. This hardship is further compounded by underdeveloped roads, seasonal tropical storms and frequent floods that burden southern Malawi [16]. Given the transient nature of these clinics, GAIA represented an ideal platform to assess the impact of an implementation of both portable pulse oximeters and IMCI continued education courses. This study aimed to understand the impact of these interventions, both individually and together, on pediatric fever diagnosis and prescribing practices in rural Malawi. While predominantly successful in these intentions, this study did conclude with some ambiguity regarding the impact of pulse oximetry as a stand-alone intervention.

## Methods

This study is a continuation of recent research regarding NMF diagnosis in Malawi which concluded that rapid point-of-care tests were needed to help providers accurately diagnose febrile patients in order to make informed decisions when prescribing antibiotics [7, 17]. Specifically, this quantitative study used a logbook review to compare provider use of the IMCI guidelines to the use of the IMCI guidelines in conjunction with pulse oximeters.

An intervention study design was employed with four comparison groups; 1) logbook review of prescribing practices from April-July 2016 prior to any interventions, referred to as control group 1; 2) logbook review of prescribing practices from April-July 2019 in clinics in which neither intervention was implemented, referred to as control group 2 to account for changes in national protocols from 2016–2019; 3) logbook review from April-July 2019 of prescribing practices at clinics receiving IMCI continued education training only; and 4) logbook review of prescribing practices from April-July 2019 at clinics receiving both IMCI continued education training and pulse oximetry training (Table 1). The data collection period took place over six weeks (May 6, 2019 –June 14, 2019). The designation of intervention versus control status to each of the five clinic sites who serviced communities in the Mulanje district was done randomly. The logbooks used for this study are filled during clinic by trained support

Table 1. Clinic sites included in each study group.

| Study Group | Clinic Site 1 | Clinic Site 2 | Clinic Site 3 | Clinic Site 4 | Clinic Site 5 |
|---|---|---|---|---|---|
| Control (2016) | X | X | X | X | X |
| Control (2019) | X | X | | | |
| Intervention 1 (IMCI-only) | | | X | | |
| Intervention 2 (IMCI+PO) | | | | X | X |

IMCI = Integrated Management of Childhood Illness, PO = pulse oximeter

staff, typically nurses or nurse aids, who verify the information written in each client's health passport and then transfer the information into the logbooks. Verification of unclear information written in the health passport is done by crosschecking with the prescribing officer to ensure that correct information is recorded. A monitoring and evaluation team at GAIA conducts data quality assessment and data verification quarterly.

Prior to the start of the study period, GAIA providers from five mobile clinics participated in one of two scheduled IMCI continued education courses held in March and April, 2019. The structure of this five-day course involved three days of theory-based learning in a classroom setting, followed by two days of practical-based learning in a pediatric clinical setting (Mulanje District Hospital). These courses were attended by 15 providers total, including both nurses and clinical officers. The participating providers were asked to take a pre-exam prior to the course, followed by a post-exam after completion of the training course. Each mobile clinic provider was assigned to a specific mobile clinic, thus the crossover of providers between mobile clinics did not occur. Clinic Site 3 was originally intended to be included in the intervention group which employed both the IMCI continued education course and the pulse oximeters. However, due to scheduling conflicts, the clinical officer from this clinic was unable to attend the pulse oximeter training, thus excluding Clinic Site 3 from the IMCI/pulse oximetry intervention group. Clinical officers and nurses from Clinic Sites 4 and 5 did participate in the training course, thus these two clinic sites comprised the IMCI/pulse oximetry intervention group.

Participating providers then attended a brief, one-hour training course in May 2019 on the use of the pulse oximeters. The training included a demonstration in the use of pulse oximeters led by the lead investigator with instruction on how to evaluate oxygen saturation levels, followed by a practical component in which the clinical staff practiced using the pulse oximeters on one another. Additional guidance was provided in the clinics by the lead investigator when needed in the weeks following the training. The training manual used for this session was drafted from the pulse oximetry training manual published by the World Health Organization, and supplemented by professional recommendations made by clinical provider members of the research team at the University of California, San Francisco [18].

The pulse oximetry protocol was designed to triage patients' respiratory status. Oxygen saturation levels between 95–100% were described as healthy, 90–95% as moderately hypoxic, and <90% as severely hypoxic, warranting immediate referral to the nearest hospital. Providers were instructed to use 95% oxygen saturation as a cut-off for antibiotic use. This threshold is based on previously established indicators of pneumonia: oxygen saturation <95%, lung crackles on auscultation, fever >37.8°C, and pulse rate >100 beats per minute [18, 19]. The providers were advised to use the 95% threshold as a general parameter to aid in their diagnostic decision-making, not as an absolute determination of a patient's diagnosis and/or need for antibiotics. Ultimately, the final diagnosis and treatments prescribed were determined by the combined knowledge gained from the pulse oximeter evaluation, IMCI guidelines, and clinical observations. Thus, for patients presenting with danger signs such as fast breathing or stridor, it was recommended that the clinical staff rely on their experience-based judgment rather than the pulse oximeter measurement as a standalone indication of health status.

The clinical staff were instructed to conduct pulse oximeter evaluation for all patients who presented with a high fever and negative malaria test. For this study, patients aged 1 year and older were evaluated with the Santamedical Generation 2 SM-165 fingertip pulse oximeter, while patients under the age of 1 year were evaluated with the Hopkins Handheld neonatal pulse oximeter [20, 21]. Both standard and neonatal pulse oximeters were distributed to the two clinics in the intervention group which received both pulse oximeters and IMCI continued education training (Clinic Sites 4 and 5) (Table 1). The lead investigator was in-country

throughout the duration of the data collection period, during which time they rotated throughout the five clinics, with particular focus on the two clinics which implemented pulse oximetry for quality assurance purposes. While both the clinical officers and nurses were trained in the use of pulse oximeters, it is primarily the role of the clinical officers to diagnose patients and prescribe medications, as such it was the clinical officers who were largely responsible for using the pulse oximeters when evaluating patients throughout this study period.

A total of 3,504 patient logbook records were included for analysis. Review of the GAIA patient logbooks allowed for assessment of the implementation of and adherence to IMCI guidelines and pulse oximeter parameters, and the resulting diagnosis and treatment of pediatric patients with NMF. A sample-size calculation was run to determine the minimum size for each group at a power of 80% and alpha of 0.05 to detect a 15% difference in practices between the four study groups, yielding a necessary 122 records per study group. Quantitative data extracted from logbooks included: clinic site, patient age, patient sex, date, diagnosis, and drugs prescribed. For the two clinics that implemented pulse oximeters, data were also recorded on whether a pulse oximeter was used and the resulting oxygen saturation measurement. To evaluate changes in the use of the pulse oximeter in the IMCI/pulse oximetry group over the six-week study period, data were collected regarding the percentage of U-5 NMF patients that were evaluated with a pulse oximeter during this time. All quantitative data were collected and stored using Redcap (version 9.1.0), a secure online platform designed for managing surveys and databases [22]. The medical records were stored in hand-written logbooks that were securely stored in the GAIA clinical offices in Mulanje and Limbe, Malawi.

After the data collection period, brief qualitative questionnaires were distributed to the GAIA clinical staff regarding their opinion of the IMCI training course and/or pulse oximeters. The questionnaire guide was drafted by the research team at the University of California, San Francisco in conjunction with GAIA administrators (S1 Text). The questionnaire was given verbally in English and recorded, with provider consent. The names of the providers were not recorded during the questionnaire, but their position as either clinical officer, nurse, or nurse aide was documented. The data collection, both logbook extraction and qualitative interviews, was conducted by the lead investigator (FS).

## Data analysis

To assess for changes in antibiotic prescriptions, the proportion of U-5 patients presenting with NMF who were prescribed antibiotics was compared among the four study groups. These proportions reflected provider adherence to either intervention when deciding to distribute or withhold antibiotics for NMF patients. To assess for differences in the prescriptive practices among the four study groups, Fisher's exact test and simple logistic regressions were used. Adherence to the IMCI guidelines was determined by assessing whether the diagnosis made by the provider was followed by the IMCI recommended treatment, as reflected in the GAIA patient logbooks. Adherence to the pulse oximetry protocol was evaluated based on the percentage of patients who received a normal pulse oximeter measurement of $\geq 95\%$ who then received the protocol's recommended treatment of basic analgesics. P-values less than 0.05 were considered significant. Odds ratios were also reported with corresponding 95% confidence intervals to provide both the strength and direction of association. All analyses were performed using the standard statistical analysis package R (version 3.5.1) [23].

The provider responses to the open-ended questionnaires were designed to help the GAIA management staff understand the experiences of the providers throughout the study and thus were considered supplemental to this study. Therefore, strict qualitative analysis methods were not employed for this portion of the study.

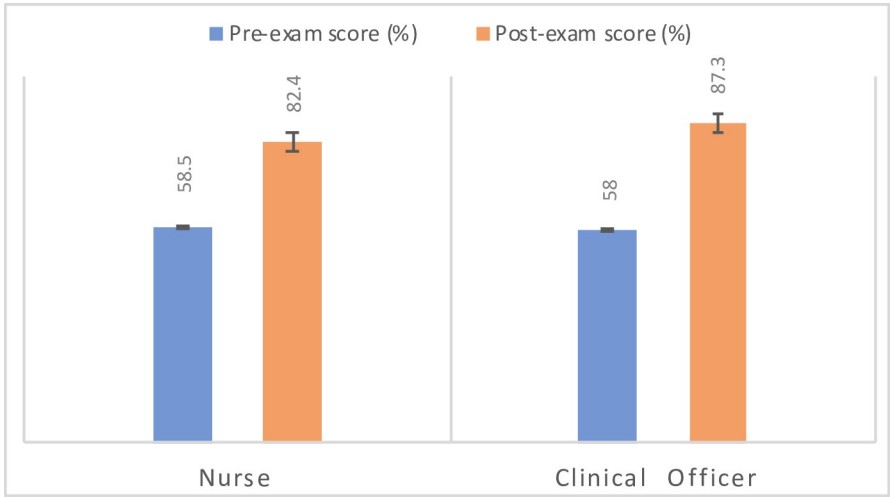

**Fig 1. Average pre- and post- exam scores for the 15 GAIA providers who participated in either the first or second IMCI continued education training course (March 4–8 or April 1–5 respectively) categorized by clinical position as either nurse or clinical officer, shown with standard deviation.**

### Ethics approval and consent to participate

All participating GAIA providers signed informed consent. The patient information collected from the logbooks was anonymized, and no direct contact between the research team and the patients took place. As such, consent was waived for this study population. Ethical approval was obtained from both the University of California San Francisco (UCSF) Committee for Human Research (#19–27452) and the Malawi National Health Sciences Research Committee (#19/03/2262) (NHSRC).

## Results

### Clinician population included in study

As previously stated, the GAIA providers who participated in the IMCI continued education course were required to complete a knowledge assessment exam before and after the course. The averages of the pre-exam and post-exam scores for the participating providers are shown in Fig 1, categorized by clinical position. Both the IMCI-only and IMCI/pulse oximetry groups had nurses and clinical officers in attendance for one of the IMCI training courses, while clinics in the 2019 control group had only nurses attend the training. A significant increase in content knowledge and understanding was detected between the pre- and post-exam scores from IMCI training (p-value <0.001). In comparing the scores of the nurses to the clinical officers, no significant difference was found in the average pre-scores, post-scores, or change in scores (p >0.001). Nurse and clinical officer participation in the pulse oximeter training is outlined in Table 2.

**Table 2. Clinical officers and nurses present for the pulse oximeter training course from each clinic site.**

|  | Clinical Officer | Nurse |
|---|---|---|
| Clinic Site 3 |  | X X X X |
| Clinic Site 4 | X | X X X |
| Clinic Site 5 | X | X X X X |

**Table 3. Demographic data for all U-5 patients presenting with NMF in each study group.**

| Variable | Control (2016) | Control (2019) | IMCI | IMCI + PO | p-value |
|---|---|---|---|---|---|
| **Sex** | | | | | |
| Male | 938 (47.9%) | 260 (45.5%) | 86 (48.3%) | 378 (47.5%) | p = 0.80 |
| Female | 1022 (52.1%) | 311 (54.5%) | 92 (51.7%) | 417 (52.5%) | |
| **Age** | | | | | |
| ≤1 month | 11 (0.6%) | 7 (1.2%) | 2 (1.1%) | 15 (1.9%) | p <0.05 |
| 2–12 months | 679 (34.6%) | 198 (34.7%) | 48 (27.0%) | 242 (30.4%) | |
| 1 | 473 (24.1%) | 164 (28.7%) | 55 (30.9%) | 211 (26.5%) | |
| 2 | 345 (17.6%) | 67 (11.7%) | 34 (19.1%) | 149 (18.7%) | |
| 3 | 258 (13.2%) | 64 (11.2%) | 19 (10.7%) | 83 (10.4%) | |
| 4 | 195 (9.9%) | 71 (12.4%) | 20 (11.2%) | 95 (11.9%) | |
| **Clinic site** | | | | | |
| 1 | 216 (11.0%) | 306 (53.5%) | – | – | |
| 2 | 410 (20.9%) | 265 (46.4%) | – | – | |
| 3 | 129 (6.6%) | – | 178 (100%) | – | |
| 4 | 643 (32.8%) | – | – | 506 (63.6%) | |
| 5 | 563 (28.7%) | – | – | 289 (36.4%) | |
| **Total participants per study group** | **1,960** | **571** | **178** | **795** | |

IMCI = Integrated Management of Childhood Illness; PO = pulse oximeter

## Patient demographics

The demographics for the patient populations included in each of the four study groups can be found in Table 3. The 2016 control data consisted of 1,960 U-5 NMF patients seen by all five clinics. The sample sizes of the three remaining study groups were significantly smaller due to division of the five clinics according to which intervention was employed. The study group with the smallest sample size was that of the IMCI-only group, which only contained data from Clinic Site 3. A significant difference in the patient age distribution was detected among the four study groups (p < 0.001).

## Differences in antibiotic prescriptive patterns

Overall, the odds of a patient receiving antibiotics in an intervention clinic that employed both IMCI training and pulse oximeters were 7.9 times less likely compared to a patient in the 2019 control group (95% CI 6.1–10.5), 7.3 times less likely than a patient in the IMCI-only group (95% CI 4.8–11.4), and 4.0 times less likely than a patient in the 2016 control group (95% CI 3.3–4.7). The significant difference in the antibiotic prescribing rates for U-5 NMF patients across the study groups can be seen in Fig 2, with rates of 75% (1,461/1,960) in 2016, 85% (485/571) in the 2019 control group, 84% (150/178) in the 2019 IMCI-only group, and 42% (336/795) in the 2019 IMCI with pulse oximetry group (p <0.001).

## Pulse oximeter utilization

Changes in the providers' use of the pulse oximeters for diagnosing U-5 NMF patients throughout the study period are shown in Fig 3. Utilization of the pulse oximeter was consistently higher in Clinic Site 5 during each week throughout the study period. However, Clinic Site 4 exhibited a steady increase in the use of the pulse oximeter over time.

A total of 795 patients were seen during the study period in intervention clinics using the pulse oximeter. Of these, 30% (n = 239) received evaluation by pulse oximetry. Differences in

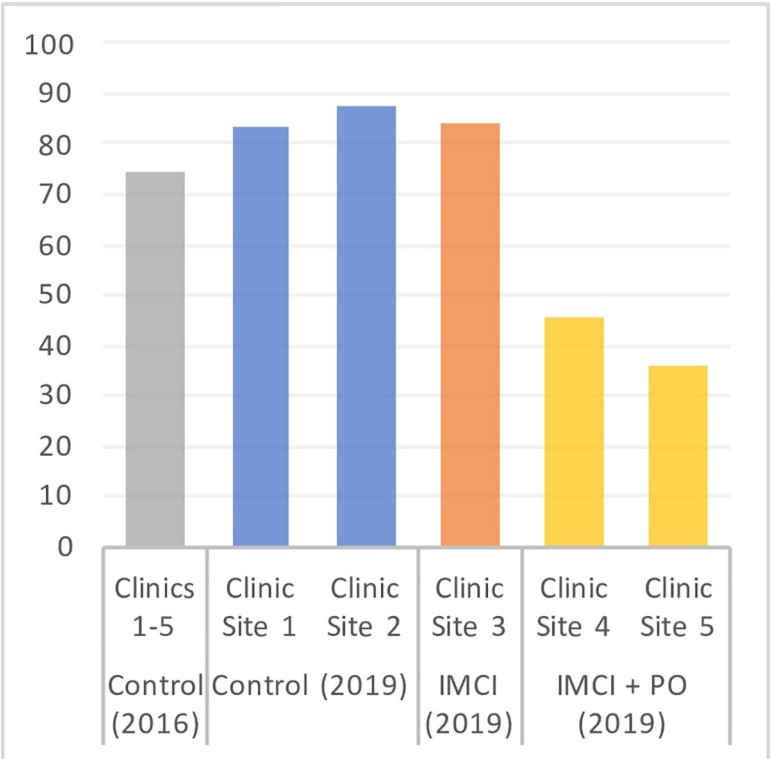

**Fig 2. Percentage of U-5 NMF patients prescribed antibiotics per clinic site within each study group: Control 2016, control 2019, IMCI-only, and IMCI/Pulse Oximeter (PO).**

patient diagnoses based on oxygen saturation cutoff can be seen in Fig 4. For patients who received pulse oximeter evaluation with a resulting oxygen saturation level greater than or equal to 95%, the most common diagnosis was common cold (31%), followed by acute respiratory infection (22%), gastroenteritis (19%), upper respiratory tract infection (18%), and sepsis (9%). Only 2% of patients with a normal oxygen saturation ($\geq$95%) were diagnosed with pneumonia. For patients who received pulse oximeter evaluation with a resulting oxygen saturation less than 95%, the most common diagnosis was pneumonia (77%), followed by acute respiratory infection (10%), sepsis (5%), bronchitis (4%), and lower respiratory tract infection (4%). A significant increase in pneumonia diagnoses was seen for pediatric patients with an oxygen saturation level less than 95% compared to patients who had an oxygen saturation of 95% or higher (p <0.001). For the remaining patients that did not receive pulse oximeter evaluation in the IMCI/pulse oximetry group, the most common diagnoses were acute respiratory infection (38%), sepsis (19%), gastroenteritis (17%), and common cold (14%). Of these patients, pneumonia represented 0.4% of all diagnoses.

## Qualitative provider interviews

From the brief qualitative interviews conducted with the GAIA providers, key themes were identified regarding the perceived benefits and challenges of implementing either intervention (Fig 5). For quotes regarding provider opinion of either intervention, see Tables 4 and 5.

   **Perceived benefits of IMCI continued education course.**   Provider responses to the questionnaires exploring the IMCI continued education course were predominantly positive. The stepwise function of the IMCI guidelines was regarded as especially important in helping diagnose

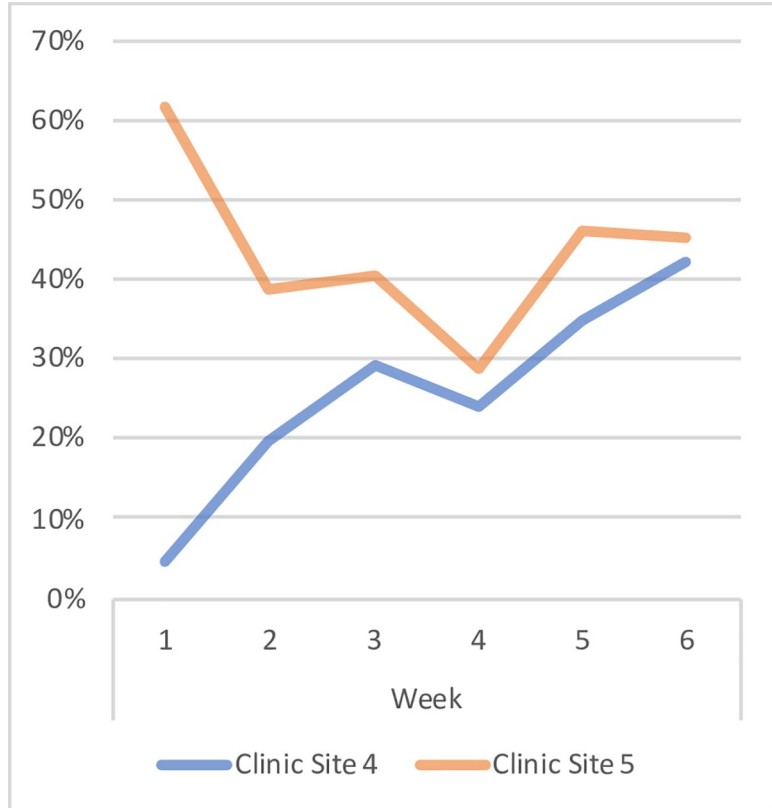

**Fig 3. Percentage of U-5 NMF patients evaluated with a pulse oximeter in clinic sites 4 and 5 over the six-week study period.**

pediatric patients. Providers felt that the enhanced classification aspect of IMCI helped them to deconstruct the symptoms presented by each patient in order to generate a holistic diagnosis.

The providers also felt they had grasped the urgency of antibiotic resistance in Malawi and the need to be more conservative with the prescription of antibiotics for febrile pediatric patients. Several providers noted the increase in antibiotic resistance that they had witnessed firsthand through working with various patient populations over the years. This experience was described as stressful, as the providers relayed feeling "frustrated" and "defeated" when children would return to clinic without any improvement from previously administered antibiotics. Lastly, many reported an augmented sense of self-confidence as a result of their participation in the IMCI training. By completing the IMCI training, many felt that they no longer had to speculate on a diagnosis.

**Perceived challenges of IMCI continued education course.** While participants did describe some challenges related to IMCI course participation, these challenges primarily involved issues regarding the scheduling of the course, rather than the content of the course itself. For future IMCI course arrangements, several providers felt that it would be beneficial to increase the duration of the course from five days to ten, with a larger portion of the training course devoted to the field-based practical component. One other criticism of the IMCI courses was that they had been scheduled in March and April, both overlapping with malaria season in Malawi. As this is the busiest season for the GAIA providers, several staff members recommended that future courses be arranged specifically in the dry season, so as to avoid major conflicts with the clinic schedules.

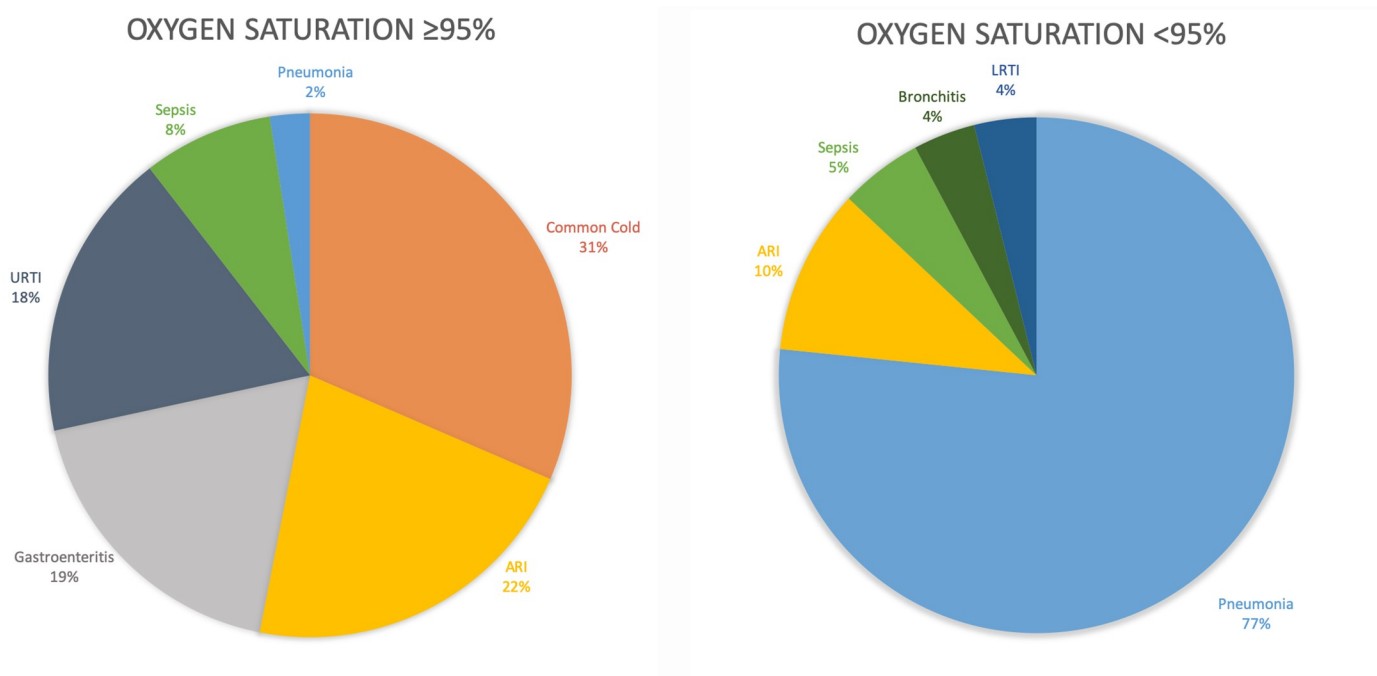

**Fig 4.** (Left) Most common diagnoses for patients that received pulse oximeter evaluation with a resulting oxygen saturation ≥95%. (Right) Most common diagnoses for patients that received pulse oximeter evaluation with a resulting oxygen saturation <95%. ARI = Acute Respiratory Infection; URTI = Upper Respiratory Infection; LRTI = Lower Respiratory Infection.

**Perceived benefits of pulse oximeter utilization.** Similar to the responses regarding IMCI training, the providers perceived the impact of implementing pulse oximeters in the mobile clinics to be beneficial. The improved detection of pneumonia as indicated by low oxygen saturation on the pulse oximeter was deemed especially useful. The clinical officers from

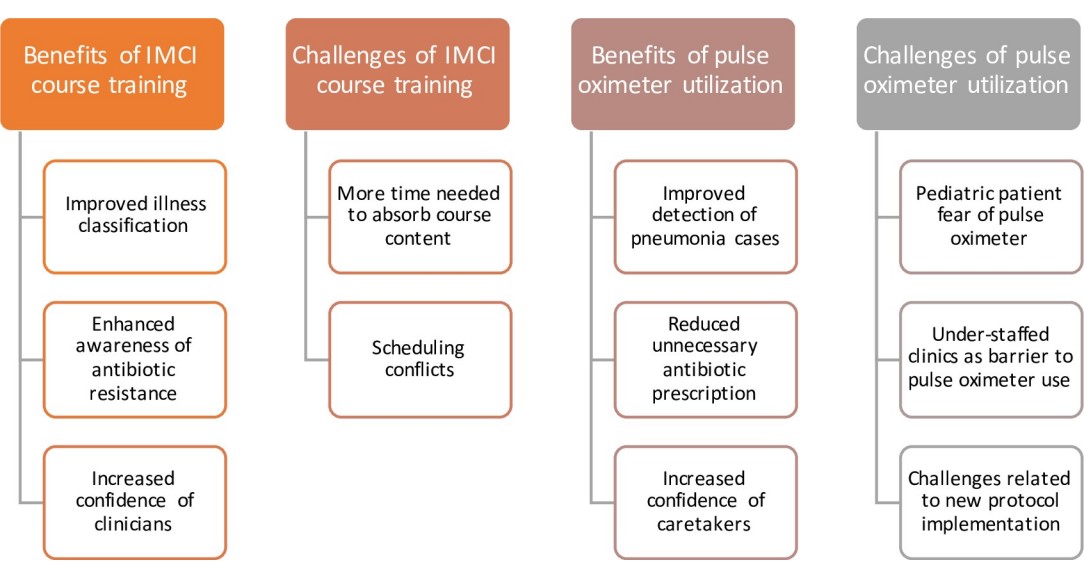

**Fig 5. Key themes from qualitative interviews held with GAIA providers regarding their opinions of the IMCI continued education course training and the pulse oximeter.**

**Table 4. Provider quotes related to their perceived benefits from IMCI course participation.**

| Benefits from IMCI course participation | Illustrative Quote |
|---|---|
| Improved illness classification | "So, if they say that the patient is coughing, we check the type of breathing the child is having. Is it fast? Is it slow? Is the child having breath weakness? Is the child having hoarse voice? Is the child sucking? If the child is not sucking, is the child eating? Is the child having diarrhea? And is the child malnourished? So, we have to check the child holistically." |
| | "IMCI it does help, especially with classification, because we do not just diagnose now. Now we classify the illness. So, we start with symptoms, we take each step to see what the symptoms are and how severe each symptom is. Then, we can come to the conclusion of what the child is suffering from." |
| Enhanced awareness of antibiotic resistance | "There is no test for antibiotic resistance. But it exists. We know that it exists because already we are being told by the Ministry of Health, and we learned in the [IMCI] training, that certain antibiotics do not work in young patients, patients under five, because resistance already exists for those antibiotics." |
| Increased provider confidence | "I think the IMCI course helped me to be more confident when I meet a patient. When I know the patient has these symptoms, I can be more certain when he does need medication, or when he just needs to rest at home to feel better. I do not feel that I am guessing about the child's condition." |

the IMCI/pulse oximetry group both believed that without the use of the pulse oximeter, many pneumonia patients may have been missed due to incomplete diagnostic evaluation. Due to the lack of diagnostic capacity in the mobile clinic setting and the rapid pace of patient flow in the mobile clinics, the providers noted the value of implementing a point-of-care device that can provide results in under 30 seconds.

In addition to providing added input when considering a diagnosis of pneumonia, the clinical officers also felt that the pulse oximeters allowed them to be more conservative when prescribing antibiotics for febrile pediatric patients. This is supported by the quantitative finding which stated an approximately 50% decrease in antibiotic prescriptive rates seen in the clinics which implemented both pulse oximeters and IMCI continued education courses. The

**Table 5. Provider quotes related to their perceived benefits of pulse oximeter utilization.**

| Benefits from pulse oximeter use | Illustrative Quote |
|---|---|
| Improved detection of pneumonia cases | "It has helped quite a lot in terms of reaching the right diagnosis, especially pneumonia, because it is very easy to miss a child with pneumonia. They can present with normal breathing, normal temperature, normal lung sounds, and so we might think it is a minor infection. We might send them home with the wrong medications. But now, we can see that their oxygen saturation is low, below 95% concentration, and we know that they do have pneumonia and do need antibiotics immediately." |
| Reduced unnecessary antibiotic prescription | "Previously, we would often prescribe antibiotics after hearing the history from the mother. We would go into the field and if we saw a child that was having difficulty breathing, he was coughing severely, and we would think maybe it would be pneumonia, we would just prescribe antibiotics. But now, when you use the pulse oximeter and see normal oxygen levels, you are indeed sure that this is not pneumonia, just an upper respiratory tract infection." |
| Increased confidence of caretakers | "Yes, the pulse oximeters have increased our confidence greatly, but also the confidence of the caregivers. You explain to them that this is medical equipment to assess if the child has pneumonia. Then, when you tell them that this is just the common cold or an upper respiratory infection, they will believe you, because you have used the instrument rather than just saying to them that their child just has the common cold. They can see the results right there and it helps them to feel that their child is safe, that they are receiving proper care." |

providers considered the 95% saturation threshold as a cutoff to which they could easily adhere, thus simplifying their decision to prescribe antibiotics versus analgesics. The use of the pulse oximeters was also believed to increase the confidence of the patients' caretakers. Many of the parents and/or guardians of the pediatric patients were reported to have felt more assured when being told that antibiotics were not needed after having observed a normal oxygen saturation reading. Witnessing the use of this technology left many caretakers with a feeling of relief in knowing that their child had been evaluated thoroughly in comparison to past clinical experiences.

**Perceived challenges of pulse oximeter implementation.** Several challenges were addressed regarding the use of the pulse oximeters in clinic. The primary challenge identified by the providers was that of applying the pulse oximeters to infants. Often times the very young children, particularly neonates and infants, demonstrated resistance to wearing the pulse oximeter. This resistance appeared to result from pediatric patients associating the use of the pulse oximeter with the use of mRDTs, and thus anticipating a painful experience. This resistance led providers to feel as though continuing to apply the pulse oximeter to the distressed child would be unethical. Additionally, one provider mentioned that on days in which the clinical officer was not working, it was difficult to incorporate the pulse oximeter into the diagnostic routine as the providers were already struggling to manage an under-staffed clinic. Lastly, several providers noted that issues related to the implementation of any new clinical protocol were observed, namely the missed documentation of the use of the pulse oximeter in the first few weeks of the study. However, these providers believed that the process of documenting pulse oximeter evaluations had improved over the six weeks.

## Discussion

We investigated the impact of implementing two interventions, pulse oximeters and IMCI continued education training, on the frequency of antibiotic prescriptions for U-5 NMF patients being treated in five mobile health clinics in Mulanje, Malawi. Data extracted from patient logbooks indicated a substantial reduction in antibiotic provision in clinics receiving both IMCI continued education training and portable pulse oximeters. However, there was no significant reduction in antibiotic prescribing practices among providers in the clinics which only implemented IMCI training. Data compiled from the provider interviews suggested additional benefits of provider participation in the IMCI training related to improved diagnostic confidence and personal empowerment.

To our knowledge, this study was one of the first to show a significant association between the use of pulse oximeters and decreased antibiotic prescriptions for febrile patients being treated in a rural mobile clinic setting. However, investigation into the impact of pulse oximeters on antibiotic prescribing practices has been expanding for urban outpatient settings in sub-Saharan Africa [24]. The rate of antibiotics prescribed in the intervention group which utilized pulse oximeters was found to be approximately half of the rate exhibited in the three remaining study groups. This outcome is consistent with previous data which found that improved identification of various causes of non-malarial fever through the use of diagnostic resources, such as point-of-care procalcitonin tests, can decrease unnecessary antibiotic consumption by 30–50% [25, 26]. This finding indicates that the relatively simple and low-cost introduction of pulse oximeters into low-resource clinics that serve rural patient communities has the potential to greatly conserve antibiotic resources. Not only can this conservation help to deter the further development of resistance, but it can also benefit clinical organizations by allowing for increased financial savings and reallocation of funds given the higher cost of antibiotics compared to basic analgesics [27, 28]. Without a dedicated follow-up component of

this study, it is impossible to declare with certainty whether this finding is purely a reduction in unnecessary antibiotic prescriptions or rather a reduction in necessary antibiotics as well. However, based on previous studies done within this region, the qualitative interviews held with the GAIA providers, and recommendations made by the Malawi Ministry of Health, it is known that the issue of antibiotic over-prescribing is quite prevalent in Malawi [7–10, 29]. Thus, it can be inferred that a more conservative shift in antibiotic prescribing practices in this setting could be beneficial in the advancement of public health.

During the qualitative interviews, the GAIA clinicians unanimously expressed greater confidence in their decision to diagnose pediatric pneumonia when presented with oxygen saturation measurements within the hypoxic range. This qualitative finding, in conjunction with the significantly higher proportion of pneumonia diagnoses for patients with a low oxygen saturation measurement, represents a meaningful clinical improvement. Given the high burden that pneumonia represents amongst pediatric patients in Malawi, and elsewhere in sub-Saharan Africa, and the ability of pulse oximeters to function in limited-resource settings, this device could greatly aid in the identification and treatment of pediatric pneumonia in rural areas [3].

It is worth noting that the IMCI-only intervention group did not exhibit a significant decrease in the number and/or percentage of U-5 NMF patients prescribed antibiotics. This finding contradicts previous studies which found that IMCI courses increased providers' adherence to guidelines related to antibiotic distribution based on patient need [30]. It should be considered, however, that this finding may be influenced by the small sample size of the IMCI-only group in comparison to the other study groups. In the majority of the provider interviews, participants discussed the benefit of the IMCI course in training them to be more conservative with antibiotic prescriptions. However, such an outcome was not reflected in the analysis. One explanation for this inconsistency could be the lack of physical confirmation provided by the IMCI guidelines compared to the objective measurements offered by the pulse oximeters. It is understandable that when a pediatric patient presents with fever, despite no other physical signs of pneumonia as outlined in the IMCI guidelines, that a provider will still prescribe antibiotics to ensure the child's safety. Furthermore, many of the guardians who accompany U-5 children may expect to leave a clinic with antibiotics, it is also likely that they strongly insist on such a prescription [31, 32]. However, it was observed that for many guardians who were able to see that their child's oxygen saturation results fell within the healthy range, they felt confident enough in this "new" technology to return home with basic analgesics.

The lack of significant change in antibiotic prescriptions within the IMCI-only study group should not undermine the additional benefits that this course provided to the providers. Throughout the interviews, several providers conveyed a sense of empowerment after having completed the IMCI course. The significant increase in knowledge retention shown through their pre- and post-exam scores indicates the educational benefit of this course, both in theory and practice. This rise in diagnostic confidence is helpful in maintaining morale, and thus should not be overlooked. Furthermore, the augmented confidence that many providers experienced after completing this course represents a meaningful achievement given the lack of resources allotted in this setting.

As mentioned above, the most significant finding of this study is that antibiotics were significantly reduced in the intervention group that received both IMCI continued education training and use of the portable pulse oximeter. Both clinics in this intervention group had a clinical officer present during the study period. In Malawi, as with many countries around the world, nurses are often viewed as inferior to clinical officers and/or physicians, despite both completing three years of healthcare education [33, 34]. As such, when the clinical officer is present, he or she exclusively determines patient diagnoses and medication prescriptions.

During the provider interviews, nurses from the IMCI-only clinic and the 2019 control clinics who had completed IMCI training expressed the need for decreasing unnecessary antibiotic prescriptions in order to deter further development of resistance. Yet, these clinics both exhibited high rates of antibiotic prescription. This finding indicates that regardless of training, it is ultimately the clinical officer who will govern significant changes in clinical output, a practice that limits the impact of positive trainings on other provider cadres. For an intervention to significantly impact diagnostic and prescriptive trends, nurses need to be given more responsibility and authority in the process of treating patients. Indeed, other studies in similar settings are beginning to show the significant impact of further empowering nurses in their clinical duties [35].

## Limitations

Due to the limited time allotted for the completion of this study, conducting patient follow-up to assess whether the diagnosis and treatment plan was appropriate was not possible. While the results from the pulse oximeter intervention group showed a significant decrease in antibiotic prescriptions, it was not possible to conclude whether this decrease in antibiotics had any impact on patient wellbeing. However, several of the collaborators on this study recently published findings which showed that 14 days after U-5 febrile patients were evaluated in the same clinical setting, outcomes were stable to improved [7, 36]. Furthermore, given the significant increase in pneumonia diagnoses seen in the IMCI/pulse oximetry group, it is more likely that the wellbeing of pediatric patients was improved given the increased detection of pneumonia cases which may otherwise have been missed. Future studies should prioritize follow-up of patient outcomes post-clinic visit.

The restriction of the IMCI-only intervention group to only one clinic site may have limited the evaluation of this intervention. Furthermore, it is possible that the significant impact of the pulse oximeter on prescriptive trends was a result of its implementation in conjunction with the IMCI training rather than as a stand-alone diagnostic intervention. Previous studies have found this combination of interventions to be optimal in improving the detection and treatment of pediatric pneumonia in resource-poor settings [37]. Future studies should also investigate the impact of these interventions by evaluating before and after prescribing practices among same clinics, rather than between clinics, to further clarify the changes that resulted directly from intervention implementation.

Finally, it is possible that clinician-level practices also affected the overall treatment patients received. However, while for the purposes of this study the clinician/clinic groupings were static based on the intervention level, in practice GAIA clinicians and/or vehicles may be on a different given route at a given time. While this helped in the establishing of the baseline from which the analysis drew, future studies could also look at the impact of clustering on changes in practice.

## Conclusion

In assessing the impact of multiple interventions on the diagnosis and treatment of pediatric non-malarial fever, this study found the use of simple pulse oximetry coupled with IMCI training significantly curbed the distribution of antibiotics in a low-resource setting. This finding points to the inclusion of pulse oximeters as a tool in the fight against antimicrobial resistance globally. Furthermore, the increase in pneumonia diagnoses in clinics using pulse oximeters indicates the benefit of this device in detecting pneumonia among pediatric patients experiencing hypoxia in similar settings. While the IMCI continued education course was not found to significantly influence antibiotic prescriptive trends, it was considered to be beneficial in

stimulating provider confidence and empowerment. Additional investigation is needed to determine whether the success of the pulse oximeter can be replicated independently of IMCI training or if the combination of the two interventions provides the ideal balance of educational and physical resources to aid in diagnosing NMF pediatric patients. Future studies are also needed to closely follow patient outcomes when antibiotics are withheld and pre-determined oxygen saturation levels are used to guide provider treatment decisions.

## Supporting information

**S1 Text. Qualitative questionnaire provided to GAIA clinical staff.**
(DOCX)

**S1 Dataset. Quantitative data extracted from logbooks.**
(XLSX)

## Acknowledgments

The authors would like to thank the Global AIDS Interfaith Alliance (GAIA), with specific gratitude extended to the in-country team members who were particularly invested in this study; Joyce Jere, Nelson Khozomba, and Mphatso Phiri. Additional appreciation is also extended to Anna Muller, Elizabeth Geoffroy, and Ellen Schell for their support in editing the final manuscript.

## Author Contributions

**Conceptualization:** Fiona Sylvies.

**Data curation:** Fiona Sylvies.

**Formal analysis:** Fiona Sylvies, Alden Blair.

**Funding acquisition:** Lucy Nyirenda, Alden Blair, Kimberly Baltzell.

**Investigation:** Fiona Sylvies.

**Methodology:** Fiona Sylvies.

**Project administration:** Lucy Nyirenda, Alden Blair, Kimberly Baltzell.

**Resources:** Lucy Nyirenda, Kimberly Baltzell.

**Software:** Alden Blair.

**Supervision:** Lucy Nyirenda, Kimberly Baltzell.

**Validation:** Lucy Nyirenda, Kimberly Baltzell.

**Visualization:** Fiona Sylvies, Alden Blair.

**Writing – original draft:** Fiona Sylvies.

**Writing – review & editing:** Fiona Sylvies, Lucy Nyirenda, Alden Blair, Kimberly Baltzell.

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
