## [Decision Letter · Decision Letter 0]

13 Jul 2020

PONE-D-20-13967

The impact of pulse oximetry and Integrated Management of Childhood Illness (IMCI) training on antibiotic prescribing practices in rural Malawi: a mixed-methods study.

PLOS ONE

Dear Dr. Sylvies,

Thank you for submitting your manuscript to PLOS ONE. After careful consideration, we feel that it has merit but does not fully meet PLOS ONE’s publication criteria as it currently stands. Therefore, we invite you to submit a revised version of the manuscript that addresses the points raised during the review process.

Please look that the reviewers have, correctly, suggested several methodological changes and clarifications to be done, and this must be observed and performed in the next version. 

We look forward to receiving your revised manuscript.

Kind regards,

Ricardo Q. Gurgel, PhD

Academic Editor

PLOS ONE

Journal Requirements:

"This study was funded by the Global AIDS Interfaith Alliance and the Institute of Global Health Sciences

 of the University of California, San Francisco. The design of this study was led by a masters student at

UCSF Institute for Global Health Sciences in collaboration with GAIA leadership."

"The authors received no specific funding for this work. "

5. Your ethics statement must appear in the Methods section of your manuscript. If your ethics statement is written in any section besides the Methods, please move it to the Methods section and delete it from any other section. Please also ensure that your ethics statement is included in your manuscript, as the ethics section of your online submission will not be published alongside your manuscript.

Reviewers' comments:

Reviewer's Responses to Questions

**Comments to the Author**

1. Is the manuscript technically sound, and do the data support the conclusions?

Reviewer #1: Yes

Reviewer #2: No

Reviewer #3: No

2. Has the statistical analysis been performed appropriately and rigorously? 

Reviewer #1: Yes

Reviewer #2: No

Reviewer #3: No

3. Have the authors made all data underlying the findings in their manuscript fully available?

Reviewer #1: Yes

Reviewer #2: No

Reviewer #3: No

4. Is the manuscript presented in an intelligible fashion and written in standard English?

Reviewer #1: Yes

Reviewer #2: No

Reviewer #3: Yes

5. Review Comments to the Author

Reviewer #1: Every day, millions of parents seek health care for their sick children, taking them to hospitals, health centres, pharmacists, doctors and traditional healers. Surveys reveal that many sick children are not properly assessed and treated by these health care providers, and that their parents are poorly advised. At first-level health facilities in low-income countries, diagnostic supports such as radiology and laboratory services are minimal or non-existent, and drugs and equipment are often scarce. Limited supplies and equipment, combined with an irregular flow of patients, leave health workers at this level with few opportunities to practice complicated clinical procedures.

These factors make providing quality care to sick children a serious challenge. WHO and UNICEF have addressed this challenge by developing a strategy called the Integrated Management of Childhood Illness (IMCI).

IMCI is an integrated approach to child health that focuses on the well-being of the whole child. IMCI aims to reduce death, illness and disability, and to promote improved growth and development among children under five years of age. IMCI includes both preventive and curative elements that are implemented by families and communities as well as by health facilities.

The strategy includes three main components: Improving case management skills of health-care staff, Improving overall health systems,Improving family and community health practices.

In health facilities, the IMCI strategy promotes the accurate identification of childhood illnesses in outpatient settings, ensures appropriate combined treatment of all major illnesses, strengthens the counselling of caretakers, and speeds up the referral of severely ill children. In the home setting, it promotes appropriate care seeking behaviours, improved nutrition and preventative care, and the correct implementation of prescribed care. So, this research is very important.

Reviewer #2: The current study which was aimed to understand the impact of two interventions, that is, IMCI continued education courses and portable pulse oximeter, both individually and together, on paediatric fever diagnosis and prescribing practices in Malawi. The authors concluded that the use of simple pulse oximetry coupled with IMCI training helped significantly curb unnecessary antibiotic prescriptions in a low-resource setting. However, this statement is not true as we don't know the outcome of these children who did not receive antibiotics compared to those who received antibiotics in IMCI + PO group. Without outcome it is not possible to say that we have reduced the “unnecessary” antibiotics. There is a likelihood that those children also needed the antibiotics.

we all know that there is no point of care diagnostic test, such as rapid diagnostic test for malaria, for pneumonia. Therefore, health care providers, using the national IMCI chart booklet which depends on clinical signs, assess, classify and treat o refer children with pneumonia. We cannot diagnose pneumonia with pulse oximeter. Pulse oximetry is used to detect hypoxaemia in children. Therefore giving it diagnostic value in pneumonia diagnosis is not correct.

Furthermore, the version of the manuscript is not suitable to be published.

Methods: It is not written well. Some specific questions are :

• Which pulse oximeters were used in the study?

• Were some or all readings of pulse oximetry?

• Please describe any supportive supervision or monitoring visits were performed for quality assurance?

• Training material for pulse oximetry?

• How many days of training on how to use pulse oximeters in children?

• Any practical training on how to use pulse oximeters in children?

• Who performed pulse oximeters?

• What was the criteria to performed pulse oximeters?

• please provide sample size calculations to see the difference between four study groups.

• Please provide more information on mobile clinics? Usual timings? Routine services? Free of charge or people need to pay for services?

• Please provide definitions of pneumonia as well as algorithm to treat or refer pneumonia cases. What about children with danger signs? Was pulse oximeter <95% is part of definition of pneumonia where IMCI + PO was the intervention?

• Line 115: the standard duration for IMCI training is of 6 days. Please give reasons why did you conduct training for 5 days?

• Line129: The rationale use for oxygen saturation as <95% for prescription of antibiotics is based on a study which studied adult population. I am not sure why did authors use this threshold level? The WHO IMCI chart booklet uses a threshold level of oxygen saturation as <90% for hypoxaemia and these children need immediate hospital referral.

• Line 135: what was the selection criteria for patient logbook records. Please explain.

• Line 138: Quantitative data extracted from logbooks included….what about sex of the child?

• Line 147: After the data collection period, brief qualitative questionnaires…..please explain more about this questionnaire, who prepare it, was it tested before use?

Results:

• Figure 1:

o caption. It seems typo error for number of providers. It should be 13, see table 2.

o Along with average scores, it seems either range or standard deviation was also presented. Please clarify this in the caption.

• Table 3:

o Regarding Age, Why were infant aged between 1 and 2 months excluded from the study?. Please give explanation in the methods section also.

o Please provide any reason why proportion of infants <1 months increased in 2019 (including control 2019, IMCI and IMCI + PO)?

o In the table denominator for calculation of percentages changes several times. Sometime it is column sum, while on the other places it is row total. Please correct this.

o Footnote: please correct Integrated management of Childhood Illness.

• Line 191 Differences in antibiotic prescription patterns. Please provide number of children received antibiotics by four study groups.

• Line 199 Pulse oximeter utilization.

o Why 30% received evaluation by pulse oximetry? Please explain.

o Please give number of children in which pulse oximetry was performed but there was no reading/missing values.

Discussion

• Line 292: no significant reduction in antibiotic prescribing….but one reason is due to small sample size in this group.

Reviewer #3: Comments

The paper tackles an important topic, one that is critical in the management of childhood illnesses. Pneumonia diagnosis in children is complex often leads to misdiagnoses and over-prescription of antibiotics. The paper evaluated the introduction of a pulse oximeter in the diagnosis of pneumonia among children under-five by trained clinicians. As appealing and relevant as the topic is, my primary concern about the paper is on the design and analysis.

1) The description of the study groups is very confusing and does not clearly separate the baseline measurement versus endline and whether the groups are comparable. Table 1 helps to understand the design better, but also shows the flaws in the analysis and comparisons that were made. The 2016 control pools data from all five sites included in the study. This is then compared to subgroups of the sites, one subgroup (sites 4 and 5) having received training on IMCI and pulse oximetry (PO), another one (site 3) received IMCI training only, and the remaining 2 (sites 1 and 2) received nothing. The pooled 2016 control group is not comparable to each individual subgroup, and the authors seem to be comparing apples and oranges. The result of 75% antibiotic prescription in the 2016 control group is not comparable to the 85% in sites 1 and 2 (2019 control), or 84% in the site 3 (IMCI only) or 42% in sites 4 & 5 (IMCI+PO). To be valid, the change must be assessed within the same group at baseline and endline. The inconsistency is further highlighted in the higher antibiotic prescription found in the 2019 control group or the IMC training only group compared to the 2016 group. The correct design and internally valid approach would be to compare changes in antibiotics prescription between baseline and endline within each site (or subgroup of sites).

2) It would be useful to also describe the profile of the clinicians and nurses that provided child care and show how they differ across sites. Figure 3 shows, for example, a progressive increase in the use of PO in one of the site that received IMCI+PO training compared to the other site that received the same training, where the use started at much higher level. It is unclear how this would be the case, but illustrates that the sites are different across.

3) It seems that nurses from the 2019 control sites have also received the IMCI only training (page 8, line 176), but this was not accounted for in the description of the results.

4) The rationale for the selection of the sites must be described.

5) The completeness of the logbooks and quality of recording must also be described and not be assumed. Any supervisory measures implemented during the study that may affect the outcomes must be acknowledged and described.

6) It is also unclear what the eligibility for the PO was. The description in the paper suggests it would fever. In the group that was trained in IMCI+PO, 30% received PO. However, among those who did not receive PO, diagnosis included symptoms that would be associated with fever as well (38% ARI, 19% sepsis). Can we assume that this group might be severely rationing the antibiotics because they were being observed?

7) Please indicate who were the data collectors, for both the extraction and the qualitative interviews. The duration of the data extraction and the number of cases extracted per logbook must also be described.

8) The positive results of the qualitative interviews in the group that received IMCI training only don’t seem to square with the high level of antibiotic prescription in that group.

9) The analysis ignored the clustering of sick children within providers. This would affect the standard errors and the inference. The logit regressions that were run were not shown. The odds ratios must be adjusted not only for this clustering effect but also for other sites and provider characteristics.

6. PLOS authors have the option to publish the peer review history of their article (what does this mean?). If published, this will include your full peer review and any attached files.

Reviewer #1: **Yes: **Ayele Mamo Abebe

Reviewer #2: **Yes: **Yasir Bin Nisar

Reviewer #3: No

---

## [Author Response · Author response to Decision Letter 0]

21 Aug 2020

Editor Comments: 

Based on the styling format required by PLOS ONE, we have made the following edits: 

• All level 1 headings have been made into bold, 18-point font, level 2 headings into bold, 16-point font, and level 3 headings into bold, 14-point font 

• We have changed all “Figure X” to “Fig X”

• We have removed all Italics from text 

• We have renamed the Figure files to “FigX.pdf”

• We have reformatted the author information to match the style of PLOS ONE standards.

• We have replaced the symbols in Tables 1 and 2 with “X”

We have included the questionnaire used in this study as supporting information, uploaded under the title “S1 Text”. As the GAIA staff members are all proficient in English speaking and comprehension, having a version of the questionnaire translated into Chichewa was deemed unnecessary. 

3. We note that you have provided funding information that is not currently declared in your Funding Statement. However, funding information should not appear in the Acknowledgments section or other areas of your manuscript. We will only publish funding information present in the Funding Statement section of the online submission form. Please remove any funding-related text from the manuscript and let us know how you would like to update your Funding Statement. Currently, your Funding Statement reads as follows:

We have removed all funding-related texted from the manuscript. Please update our Funding Statement to the following: "This study was funded by the Global AIDS Interfaith Alliance and the Institute of Global Health Sciences of the University of California, San Francisco. The authors received no specific funding for this work."

After speaking with our in-country team, we have been granted permission to publish our dataset publicly without specific names of clinics and/or clinicians involved. We have included this dataset as Supporting Information entitled “S2 Dataset”. We have updated this information in line 548 under “Availability of data and materials”.

5. Your ethics statement must appear in the Methods section of your manuscript. If your ethics statement is written in any section besides the Methods, please move it to the Methods section and delete it from any other section. Please also ensure that your ethics statement is included in your manuscript, as the ethics section of your online submission will not be published alongside your manuscript.

We moved the section entitled “Ethics Approval and Consent to Participate” to the final paragraph of the Methods section. 

Reviewer 1:

Thank you so much for your feedback, we greatly appreciate your support of our research and hope that our findings may contribute to future efforts to improve community health in low resource settings. 

Reviewer 2: 

Overall:

1. The authors concluded that the use of simple pulse oximetry coupled with IMCI training helped significantly curb unnecessary antibiotic prescriptions in a low-resource setting. However, this statement is not true as we don't know the outcome of these children who did not receive antibiotics compared to those who received antibiotics in IMCI + PO group. Without outcome it is not possible to say that we have reduced the “unnecessary” antibiotics. There is a likelihood that those children also needed the antibiotics.

We very much appreciate your input. Based on our literature search and conversations with the in-country clinicians, we found that the issue of inappropriate antibiotic prescription was very prevalent in this region of Malawi and needed to be addressed. However, to your point, we have made the following edits throughout the paper: 

• “Without a dedicated follow-up component of this study, it is impossible to declare with certainty whether this finding is purely a reduction in unnecessary antibiotic prescriptions or rather a reduction in necessary antibiotics as well. However, based on previous studies done within this region, the qualitative interviews held with the GAIA providers, and recommendations made by the Malawi Ministry of Health, it is known that the issue of antibiotic over-prescribing is quite prevalent in Malawi. Thus, it can be inferred that a more conservative shift in antibiotic prescribing practices in this setting could be beneficial in the advancement of public health.”

• We have also addressed this issue in the limitations section of this paper: “Due to the limited time allotted for the completion of this study, conducting patient follow-up to assess whether the diagnosis and treatment plan was appropriate was not possible. While the results from the pulse oximeter intervention group showed a significant decrease in antibiotic prescriptions, it was not possible to conclude whether this decrease in antibiotics had any impact on patient wellbeing. However, several of the collaborators on this study recently published findings which showed that 14 days after U-5 febrile patients were evaluated in the same clinical setting, outcomes were stable to improved.(7,34) Furthermore, given the significant increase in pneumonia diagnoses seen in the IMCI/pulse oximetry group, it is more likely that the wellbeing of pediatric patients was improved given the increased detection of pneumonia cases which may otherwise have been missed. Future studies should prioritize follow-up of patient outcomes post-clinic visit.”

2. We all know that there is no point of care diagnostic test, such as rapid diagnostic test for malaria, for pneumonia. Therefore, health care providers, using the national IMCI chart booklet which depends on clinical signs, assess, classify and treat or refer children with pneumonia. We cannot diagnose pneumonia with pulse oximeter. Pulse oximetry is used to detect hypoxaemia in children. Therefore giving it diagnostic value in pneumonia diagnosis is not correct.

We do apologize if our wording suggested the pulse oximeter as a definitive diagnostic tool for pneumonia. Our aim was to introduce the pulse oximeters as a tool that could provide useful information for the clinician’s diagnostic decision-making, in addition to their knowledge gained from the IMCI training and clinical expertise. However, we have added the following statements throughout the paper in order to clarify this point: 

• In the abstract, we removed the line, “Additionally, the pulse oximeters demonstrated the capacity to improve detection of pediatric pneumonia” and replaced it with, “Enhanced detection of hypoxaemia in pediatric patients was regarded by clinicians as helpful for identifying pneumonia cases.”

• “The providers were advised to use the 95% threshold as a general parameter to aid in their diagnostic decision-making, not as an absolute determination of a patient’s diagnosis and/or need for antibiotics. Ultimately, the final diagnosis and treatments prescribed were determined by the combined knowledge gained from the pulse oximeter evaluation, IMCI guidelines, and clinical observations. Thus, for patients presenting with danger signs such as fast breathing or stridor, it was recommended that the clinical staff rely on their experience-based judgment rather than the pulse oximeter measurement as a standalone indication of health status.” Line 156

• In the discussion, we removed the following statement, “Of the various benefits of utilizing pulse oximeters in clinic, the subsequent improvement in pneumonia detection was universally commended by providers during the interviews.” This line was replaced with: “During the qualitative interviews, the GAIA clinicians unanimously expressed greater confidence in their decision to diagnose pediatric pneumonia when presented with oxygen saturation measurements within the hypoxic range.” Line 443

Methods: 

1. Which pulse oximeters were used in the study?

 For patients aged approximately 1 year and older, the Santamedical Generation 2 SM-165 Fingertip Pulse Oximeter was used, while patients under the age of 1 year were evaluated with the Hopkins Handheld Pulse Oximeter. We have included this information in the methods section in line 168. 

2. Please describe any supportive supervision or monitoring visits were performed for quality assurance?

The lead investigator was in-country throughout the duration of the data collection period, during which time they rotated throughout the 5 clinics, with particular focus on the 2 clinics which implemented pulse oximetry for quality assurance purposes. This information has been added to the methods section to line 173. 

3. Training material for pulse oximetry?

The training schedule for pulse oximetry was drafted based on both existing training resources found online, primarily the Pulse Oximetry Training Manual published by the World Health Organization in 2011, as well as the professional recommendations made by clinical provider members of the study team. This information has been added to the methods section to line 147.

4. How many days of training on how to use pulse oximeters in children?

The training course was brief, lasting 1 hour. However, additional guidance was provided for the clinicians assigned to clinics which received this intervention in the weeks following the training, although very little guidance was needed. This information has been added to the methods section to line 142.

5. Any practical training on how to use pulse oximeters in children?

The training session began with a demonstration in the use of pulse oximeters with instructions on how to interpret the resulting oxygen saturation levels, followed by a practical component in which the clinical staff practiced using the pulse oximeters on one another. There was no specific training session in which children were present, however guidance was offered by the lead researcher in clinic when needed, particularly with the neonatal pulse oximeters. This information has been added to the methods section to line 143.

6. Who performed pulse oximeters?

While both the clinical officers and nurses were trained in the use of pulse oximeters, it is primarily the role of the clinical officers to diagnose patients and prescribe medications, as such it was the clinical officers who were largely responsible for using the pulse oximeters when evaluating patients throughout this study. This information has been added to the methods section to line 175. 

7. What was the criteria to performed pulse oximeters?

When training the clinical staff in the use of pulse oximeters, they were instructed to evaluate all patients who presented with a high fever and negative malaria test. This information has been added to the methods section to line 167. 

8. Please provide sample size calculations to see the difference between four study groups.

We thank the reviewer for bringing up the important issues of whether or not we had an accurate sample size and a reasonable power to detect meaningful differences between our groups. As we did detect multiple significant differences between the groups, we omitted the calculations from our paper for space, but now recognize that many readers would still wish to see it. We have therefore added the following sentence within our methods section:

"A sample-size calculate was run to determine the minimum size for each group at a power of 80% and alpha of 0.05 to detect a 15% difference in practices between the four study groups, yielding a necessary 122 records per study group." Line 182

9. Please provide more information on mobile clinics? Usual timings? Routine services? Free of charge or people need to pay for services?

The GAIA clinics operate five days a week, often extending into the weekend, offering comprehensive primary care services and consistent treatment for conditions that heavily plague this region such as malaria, tuberculosis, and HIV, all free of charge for their patients. This information has been added to the background section to line 92. 

10. Please provide definitions of pneumonia as well as algorithm to treat or refer pneumonia cases. What about children with danger signs? Was pulse oximeter <95% is part of definition of pneumonia where IMCI + PO was the intervention?

We added the following statements to the background and methods sections for clarification: 

• “Defined as an inflammatory infection of the alveoli in the lungs, pneumonia typically presents with cough, fatigue, chest tightness, fever, sweating, and shortness of breath.” Line 62. 

• “For patients presenting with danger signs such as fast breathing or stridor, it was recommended that the clinical staff rely on their experience-based judgment rather than the pulse oximeter measurement as a standalone indication of health status.” Line 158. 

• While saturations measurements <95% were not defined as diagnostic of pneumonia, the clinical staff were instructed on the association between pneumonia and moderate hypoxia and thus used the 95% threshold as a tool to help determine whether a patient’s condition warranted antibiotic use or simply analgesics. Ultimately, the decision to diagnose pneumonia and/or prescribe antibiotics was based off of the clinician’s whole evaluation of the patient, including what they learned from the IMCI training and observed of the patient’s physical presentation. This has been further clarified in the methods section in line 156. 

11. Line 115: the standard duration for IMCI training is of 6 days. Please give reasons why did you conduct training for 5 days?

The IMCI training was led by an experienced team based in Malawi. When presented with the question of why they conducted the training in 5 days rather than 6, they responded: “The extra (6th) day is a provision to ensure that all participants have completed a number of practices. In this case, there were enough clinical cases and additional onsite facilitators during the training days which enabled all participants to complete within 5 days.” 

12. Line129: The rationale use for oxygen saturation as <95% for prescription of antibiotics is based on a study which studied adult population. I am not sure why did authors use this threshold level? The WHO IMCI chart booklet uses a threshold level of oxygen saturation as <90% for hypoxaemia and these children need immediate hospital referral.

Our rationale for using the 95% cutoff instead of 90% was that it would be more conservative, and thus a safer approach to treating pediatric patients. By making the cutoff 95%, more children would be receiving antibiotics comparatively and would thus minimize risk. We have expanded on this point for clarification: 

• “The pulse oximetry protocol was designed to triage patients’ respiratory status. Oxygen saturation levels between 95-100% were described as healthy, 90-95% as moderately hypoxic, and <90% as severely hypoxic, warranting immediate referral to the nearest hospital.” Line 151. 

13. Line 135: what was the selection criteria for patient logbook records. Please explain.

Patient logbook records were included based on these inclusion criteria: age under five years, febrile, malaria-negative, and treated during the dry season, as stated in the methods section in line 33. 

14. Line 138: Quantitative data extracted from logbooks included….what about sex of the child?

Thank you for catching this error. We have added patient sex to the list of data extracted from the logbooks in line 185 of the methods section. 

15. Line 147: After the data collection period, brief qualitative questionnaires…..please explain more about this questionnaire, who prepare it, was it tested before use?

The questionnaire guide was drafted by the research team at the University of California, San Francisco in conjunction with GAIA administrators. This information has been added to the methods section in line 197. 

Results:

1. Figure 1:

caption. It seems typo error for number of providers. It should be 13, see table 2.

This is not a typo, as Figure 1 is referring to the 15 GAIA clinicians who participated in the IMCI study while Table 2 is referring to the 13 GAIA clinicians who participated in the pulse oximetry training. 

Along with average scores, it seems either range or standard deviation was also presented. Please clarify this in the caption.

We have clarified in the Figure 1 caption that these bars represent the standard deviation values. 

2. Table 3:

Regarding Age, Why were infant aged between 1 and 2 months excluded from the study?. Please give explanation in the methods section also.

There was no exclusion of infants aged between 1 and 2, we have changed the category in Table 3 to read as “�1 month” instead of “<1 month” for clarification. 

Please provide any reason why proportion of infants <1 months increased in 2019 (including control 2019, IMCI and IMCI + PO)?

While we are also interested in this shift in patient age between 2016-2019, we do not believe it is within the scope of this study to fully investigate this demographic trend. 

In the table denominator for calculation of percentages changes several times. Sometime it is column sum, while on the other places it is row total. Please correct this.

We have removed the row sums from Table 3 so that only column sums remain. 

Footnote: please correct Integrated management of Childhood Illness.

Thank you for catching this error, we have corrected it. 

3. Line 191 Differences in antibiotic prescription patterns. Please provide number of children received antibiotics by four study groups.

We have included these numbers in our results section in line 309: “The significant difference in the antibiotic prescribing rates for U-5 NMF patients across the study groups can be seen in Figure 2, with rates of 75% (1,461/1,960) in 2016, 85% (485/571) in the 2019 control group, 84% (150/178) in the 2019 IMCI-only group, and 42% (336/795) in the 2019 IMCI with pulse oximetry group (p <0.001).” 

4. Line 199 Pulse oximeter utilization.

Why 30% received evaluation by pulse oximetry? Please explain.

The clinical staff did not use the pulse oximeters for 100% of the pediatric patients presenting with non-malarial fever. There were many instances in which the devices were not used either due to clinician confidence in the patient’s diagnosis without needing to know the oxygen saturation or due to lack of available time given the extremely fast pace of patient flow in these mobile clinics. 

Please give number of children in which pulse oximetry was performed but there was no reading/missing values.

To our knowledge, there were no instances in which pulse oximetry was performed without documentation of the resulting values. 

Discussion: 

1. Line 292: no significant reduction in antibiotic prescribing….but one reason is due to small sample size in this group.

We have edited the discussion to include the following statement: “It is worth noting that the IMCI-only intervention group did not exhibit a significant decrease in the number and/or percentage of U-5 NMF patients prescribed antibiotics. This finding contradicts previous studies which found that IMCI courses increased providers’ adherence to guidelines related to antibiotic distribution based on patient need. It should be noted, however, that this finding may be influenced by the small sample size of the IMCI-only group in comparison to the other study groups.” 

Reviewer #3: Comments

1) The description of the study groups is very confusing and does not clearly separate the baseline measurement versus endline and whether the groups are comparable. Table 1 helps to understand the design better, but also shows the flaws in the analysis and comparisons that were made. The 2016 control pools data from all five sites included in the study. This is then compared to subgroups of the sites, one subgroup (sites 4 and 5) having received training on IMCI and pulse oximetry (PO), another one (site 3) received IMCI training only, and the remaining 2 (sites 1 and 2) received nothing. The pooled 2016 control group is not comparable to each individual subgroup, and the authors seem to be comparing apples and oranges. The result of 75% antibiotic prescription in the 2016 control group is not comparable to the 85% in sites 1 and 2 (2019 control), or 84% in the site 3 (IMCI only) or 42% in sites 4 & 5 (IMCI+PO). To be valid, the change must be assessed within the same group at baseline and endline. The inconsistency is further highlighted in the higher antibiotic prescription found in the 2019 control group or the IMC training only group compared to the 2016 group. The correct design and internally valid approach would be to compare changes in antibiotics prescription between baseline and endline within each site (or subgroup of sites).

Since the GAIA’s initiation, each clinic’s circumstances have been comparable in regard to topography, patient demographics and socioeconomic status, health conditions treated, clinical training, and prescribing practices. Therefore, we felt pooling information across all GAIA clinics was informative. Given that individual patient data from 2016 has been converted into an amalgamated dataset, we cannot know for certain whether different providers from the 2016 group showed varying prescribing practices. In the discussion we have noted that it is possible that the individual practitioner approaches from 2016 may have influenced antibiotics prescribed. As such, future studies which evaluate the before and after prescribing practices among same clinical groups would provide further clarity for this area of study.

2) It would be useful to also describe the profile of the clinicians and nurses that provided child care and show how they differ across sites. Figure 3 shows, for example, a progressive increase in the use of PO in one of the site that received IMCI+PO training compared to the other site that received the same training, where the use started at much higher level. It is unclear how this would be the case, but illustrates that the sites are different across.

Based on our observations from the clinic, we believe that the difference in pulse oximeter utilization between clinic sites was largely due to differences in each clinical officer’s priorities. For example, the clinic that exhibited higher PO utilization was led by a clinical officer who had substantially more administrative responsibility and was thus more enthusiastic about antibiotic conservation from a financial standpoint. In the clinic which showed a more hesitant use of the pulse oximeters, the lead clinical officer was more less fiscally focused and was reluctant to withhold antibiotics from potentially sick children. While we agree that this personal information is contextually useful, we believe it to be too speculative for the purposes of this study. Furthermore, it could allow for the identification of specific GAIA clinicians, which would be inappropriate. As such, we have decided to exclude these details. 

3) It seems that nurses from the 2019 control sites have also received the IMCI only training (page 8, line 176), but this was not accounted for in the description of the results.

The nurse participation from the 2019 control group is accounted for at the beginning of the results section in line 232: “Both the IMCI-only and IMCI/pulse oximetry groups had nurses and clinical officers in attendance for one of the IMCI training courses, while clinics in the 2019 control group had only nurses attend the training.”

4) The rationale for the selection of the sites must be described.

The designation of intervention versus control status to each of the five clinic sites who serviced communities in the Mulanje district was done so randomly. This information has been added under the methods section in line 119. 

5) The completeness of the logbooks and quality of recording must also be described and not be assumed. Any supervisory measures implemented during the study that may affect the outcomes must be acknowledged and described.

The logbooks are filled during clinic by trained support staff, typically nurses or nurse aids, who verify the information written in each client’s health passport and then transfer that information into the logbooks. Verification of unclear information written in the health passport is done by crosschecking with the prescribing officer to ensure that correct information is recorded. A monitoring and evaluation team at GAIA conduct data quality assessment and data verification quarterly. This information has been added to the methods section in line 125. 

6) It is also unclear what the eligibility for the PO was. The description in the paper suggests it would fever. In the group that was trained in IMCI+PO, 30% received PO. However, among those who did not receive PO, diagnosis included symptoms that would be associated with fever as well (38% ARI, 19% sepsis). Can we assume that this group might be severely rationing the antibiotics because they were being observed?

While it is possible that the clinical officers changed their prescribing habits due to feeling observed by our research team, a similar decrease in antibiotic prescriptions was not observed for the clinics without pulse oximeter implementation during the days in which a member of the research team was present. As such, we do not believe that our presence had a significant influence over the providers’ decision to prescribe antibiotics. 

7) Please indicate who were the data collectors, for both the extraction and the qualitative interviews. The duration of the data extraction and the number of cases extracted per logbook must also be described.

The data collection, both logbook extraction and qualitative interviews, was conducted by the lead investigator (FS). This information has been added to the methods section in line 202. The duration of data extraction is referenced in the methods section in line 118: “The data collection period took place over six weeks (May 6, 2019 – June 14, 2019).” 

Unfortunately, it would be very difficult to calculate how many cases were extracted per logbook as the logbooks are kept as hand-written records and stored in the GAIA office in Mulanje. 

8) The positive results of the qualitative interviews in the group that received IMCI training only don’t seem to square with the high level of antibiotic prescription in that group.

Yes, we discussed this contradiction in the discussion in line 455: “In the majority of the provider interviews, participants discussed the benefit of the IMCI course in training them to be more conservative with antibiotic prescriptions. However, such an outcome was not reflected in the analysis. One explanation for this inconsistency could be the lack of physical confirmation provided by the IMCI guidelines compared to the objective measurements offered by the pulse oximeters. It is understandable that when a pediatric patient presents with fever, despite no other physical signs of pneumonia as outlined in the IMCI guidelines, that a provider will still prescribe antibiotics to ensure the child’s safety. Furthermore, many of the guardians who accompany U-5 children may expect to leave a clinic with antibiotics, it is also likely that they strongly insist on such a prescription.” 

9) The analysis ignored the clustering of sick children within providers. This would affect the standard errors and the inference. The logit regressions that were run were not shown. The odds ratios must be adjusted not only for this clustering effect but also for other sites and provider characteristics.

We thank the reviewer for their keen statistical considerations with regards to the analysis and issues of clustering. As we consider prescribing practices, there is certainly the possibility that the approaches used by different providers may affect the overall differences seen between groups as might characteristics of the patient population served. In these cases, it can be advantageous to cluster the analysis by facilities and/or providers themselves though a generalized estimating equation (GEE) or logistic mixed-effect model to provide a more accurate representation of the differences seen by an intervention. However, in the case of our study we ultimately decided against this for the reasons listed below:

• With regards to the patient populations: the district served by GAIA mobile clinics is a rural district of Malawi with a largely homogeneous population along the mobile clinic routes including that of demographic-, socioeconomic-, and environmental-profiles. Further, given the remote nature of the district, patients will often be as close to one mobile clinic route as to another and choose to attend one based on need and availability. 

• In contrast to static facilities with dedicated staff, the nature of the mobile clinics made clustering problematic. While the pulse-oximeter intervention was given to a set group of providers for purposes of this study, in general the staffing of the mobile clinics shifts among the GAIA staff. Further, given groups of staff and vehicles are not always serving the same area. For this reason, we could not cluster the clinics when comparing pre- and post- intervention as such clustering would be artificial and not account for providers or patient characteristics that clustering seeks to address. 

Recognizing the importance of how things like provider approaches could affect prescription rates, and the theoretical importance of clustering to account for this, we have added the following notes in the methods and discussion:

• It is possible that clinician-level practices also affected the overall treatment patients received. However, while for the purposes of this study the clinician/clinic groupings were static based on the intervention level, in practice GAIA clinicians and/or vehicles may be on a different given route at a given time. While this helped in the establishing of the baseline from which the analysis drew, future studies could also look at the impact of clustering on changes in practice. Line 517.

---

## [Decision Letter · Decision Letter 1]

3 Nov 2020

The impact of pulse oximetry and Integrated Management of Childhood Illness (IMCI) training on antibiotic prescribing practices in rural Malawi: a mixed-methods study.

PONE-D-20-13967R1

Dear Dr. Sylvies,

We’re pleased to inform you that your manuscript has been judged scientifically suitable for publication and will be formally accepted for publication once it meets all outstanding technical requirements.

Kind regards,

Ricardo Q. Gurgel, PhD

Academic Editor

PLOS ONE

Additional Editor Comments (optional):

Reviewers' comments:

Reviewer's Responses to Questions

**Comments to the Author**

1. If the authors have adequately addressed your comments raised in a previous round of review and you feel that this manuscript is now acceptable for publication, you may indicate that here to bypass the “Comments to the Author” section, enter your conflict of interest statement in the “Confidential to Editor” section, and submit your "Accept" recommendation.

Reviewer #1: (No Response)

Reviewer #2: All comments have been addressed

2. Is the manuscript technically sound, and do the data support the conclusions?

Reviewer #1: Yes

Reviewer #2: Yes

3. Has the statistical analysis been performed appropriately and rigorously? 

Reviewer #1: Yes

Reviewer #2: Yes

4. Have the authors made all data underlying the findings in their manuscript fully available?

Reviewer #1: Yes

Reviewer #2: No

5. Is the manuscript presented in an intelligible fashion and written in standard English?

Reviewer #1: Yes

Reviewer #2: Yes

6. Review Comments to the Author

Reviewer #1: Thank for the opportunity to review 'The impact of pulse oximetry and Integrated Management of Childhood Illness (IMCI) training on antibiotic prescribing practices in rural Malawi: a mixed-methods study' I have no further comments for the authors

Reviewer #2: (No Response)

7. PLOS authors have the option to publish the peer review history of their article (what does this mean?). If published, this will include your full peer review and any attached files.

Reviewer #1: **Yes: **Ayele Mamo Abebe

Reviewer #2: No

---

## [Editor Report · Acceptance letter]

9 Nov 2020

PONE-D-20-13967R1 

The impact of pulse oximetry and Integrated Management of Childhood Illness (IMCI) training on antibiotic prescribing practices in rural Malawi: a mixed-methods study. 

Dear Dr. Sylvies:

I'm pleased to inform you that your manuscript has been deemed suitable for publication in PLOS ONE. Congratulations! Your manuscript is now with our production department. 

Kind regards, 

on behalf of

Professor Ricardo Q. Gurgel 

Academic Editor

PLOS ONE